# Assessment of Vineyard Canopy Characteristics from Vigour Maps Obtained Using UAV and Satellite Imagery

**DOI:** 10.3390/s21072363

**Published:** 2021-03-29

**Authors:** Javier Campos, Francisco García-Ruíz, Emilio Gil

**Affiliations:** Department of Agro Food Engineering and Biotechnology, Universitat Politècnica de Catalunya, Esteve Terradas, 8, 08860 Castelldefels, Spain; javier.campos@upc.edu (J.C.); fco.jose.garcia@upc.edu (F.G.-R.)

**Keywords:** vineyard, pesticide application, variable rate application, unmanned aerial vehicle, satellite, nanosatellite

## Abstract

Canopy characterisation is a key factor for the success and efficiency of the pesticide application process in vineyards. Canopy measurements to determine the optimal volume rate are currently conducted manually, which is time-consuming and limits the adoption of precise methods for volume rate selection. Therefore, automated methods for canopy characterisation must be established using a rapid and reliable technology capable of providing precise information about crop structure. This research providedregression models for obtaining canopy characteristics of vineyards from unmanned aerial vehicle (UAV) and satellite images collected in three significant growth stages. Between 2018 and 2019, a total of 1400 vines were characterised manually and remotely using a UAV and a satellite-based technology. The information collected from the sampled vines was analysed by two different procedures. First, a linear relationship between the manual and remote sensing data was investigated considering every single vine as a data point. Second, the vines were clustered based on three vigour levels in the parcel, and regression models were fitted to the average values of the ground-based and remote sensing-estimated canopy parameters. Remote sensing could detect the changes in canopy characteristics associated with vegetation growth. The combination of normalised differential vegetation index (NDVI) and projected area extracted from the UAV images is correlated with the tree row volume (TRV) when raw point data were used. This relationship was improved and extended to canopy height, width, leaf wall area, and TRV when the data were clustered. Similarly, satellite-based NDVI yielded moderate coefficients of determination for canopy width with raw point data, and for canopy width, height, and TRV when the vines were clustered according to the vigour. The proposed approach should facilitate the estimation of canopy characteristics in each area of a field using a cost-effective, simple, and reliable technology, allowing variable rate application in vineyards.

## 1. Introduction

The European Green Deal, recently launched by the European Commission [1], is designed to deal with climate and environment-related challenges, attempting to develop sustainable responses. Among the several topics included in the Green Deal, agricultural activities and all aspects related to food production are addressed in the European Farm to Fork Strategy to ensure a reasonable, healthy, and environment-friendly food system. This strategy includes the impacting measures linked with the use of a plant protection product (PPP), given its negative effects on air and water quality, soil degradation, food safety, and human health.

One of the most important challenges considered in the Farm to Fork strategy is the objective to reduce the overall use and risk of chemical pesticides by 50% and the use of more hazardous pesticides by 50% by 2030. This objective is particularly important for orchard fruits and vineyards. Vineyards, while accounting for only 7% of the agricultural land area in the European Union consume 48% of the total active ingredients [2]. The crops at fruit orchards and vineyards, similar to all ’three-dimensional’ (3D) crops, are characterised by the variation in their canopy characteristics and the heterogeneity in a parcel [3], making it difficult to achieve safe and optimal pesticide application.

The latest improvements in the available technology and its adaptation to these types of crops have resulted in remarkable achievements in both the reduction in the total amount of PPP and an increased control of the losses and, consequently, in the reduction in environmental contamination. These advancements involved the use of an accurate method to identify, characterise, and quantify the amount of pesticide to be sprayed, which is considered as the most important factor related to the success of the pesticide application process. Miranda-Fuentes et al. [4] demonstrated the effects of different methods for crown characterisation in isolated olive trees on the obtained results, concluding that irrespective of the selected method for canopy evaluation, some minimum requirements in terms of accuracy must be ensured to apply the most suitable amount of pesticide. Pesticide dose and dose expression were demonstrated as two factors affected by the canopy characteristics of citrus plantations [5], showing that a large vegetation implies major differences in the canopy deposition and coverage. Drift values in spray application in apple plantations were also directly related to crop foliage characteristics [6], with a fully foliated canopy resulting in a 25-times less drift than the one obtained in a dormant canopy stage. A similar conclusion was drawn by Grella et al. [7], who showed that the crop canopy structure plays a role in determining the drift values at both apple and vineyard plantations, particularly focusing on the crop type, training system, and growth stage.

Canopy characterisation at fruit and vineyard plantations has been commonly discussed in recent years, with numerous proposed methodologies ranging from simple manual processes [8,9] to those using sophisticated devices, such as LiDAR [4,10,11], ultrasonic sensors [12,13,14], unmanned aerial vehicles (UAVs) [15,16,17], and satellites [18,19,20]. The advantages and disadvantages of all methodologies have been probed, making it difficult to select the most accurate one.

UAVs and satellite imagery, classified as remote sensing methodologies, have been promoted in the last few years [21,22] as remarkable techniques for canopy characterisation. Differences in their management, accuracy, economical cost, and other important factors have been extensively discussed, yielding various advantages and disadvantages—directly related to the targeted crop and conditions—of both methods. Although satellite image acquisition of large areas saves considerable time, it has a low and inadequate resolution for precision viticulture [23,24]. The effectiveness of Sentinel-2 imagery and high-resolution UAV aerial images was evaluated [25], concluding that the resolution of the satellite imagery was insufficient for their direct use for describing vineyard variability. In contrast, Di Gennaro et al. [26] demonstrated the effectiveness and high resolution of Sentinel-2 imagery in the canopy characterisation process at vineyard plantations. Recent advancements in UAV-related research have led to a wide range of UAV applications for monitoring vineyard performance, such as rate of canopy development, canopy structure spatial variability, and disease incidence [27,28,29,30,31]. Similar to manned aircraft and satellite-based remote sensing, UAVs are convenient in terms of simple flight preparation and flexible operations [25], independently of their technical specifications (fixed-wing or multirotor); however, they are more effective for small and medium-sized vineyards [32,33]. According to Ouyang et al. [34], the operational flexibility of UAVs allows the timely assessment of canopy management outcomes, compared with manned aircraft and satellite remote sensing.

The variable rate application (VRA) of a PPP in 3D crops represents an important step forward in the sustainable use of pesticides, allowing accurate spray deposition and reduction in the drift loss by adjusting the optimal amount of the PPP applied to the canopy structure. This technology can be implemented using two different methods. The first one is adjusting the working parameters of the spray process based on the canopy characteristics measured ‘on the go’ using electronic devices [35,36,37,38]. The second is using previously generated canopy maps by manual or remote sensing measurements and their transformation into prescription maps using dedicated tools [39,40,41]. The second option based on canopy maps requires the implementation of an accurate process for canopy characterisation, and its extension and implementation on a large scale in commercial parcels are directly related to the degree of automation and ease of the process [42], being the development of canopy maps the most influencing process.

There is a need to develop an automated method for canopy characterisation that can consequently promote the implementation of the VRA process for sustainable PPP management in vineyards. Other studies on canopy characterisation in orchards and fruit crops have been conducted based on the use of the normalised differential vegetation index (NDVI) as the main parameter obtained utilising remote sensing platforms and its relationship with principal canopy dimensions [43].

The general objective of this research was to investigate the potential relationships among manual field measurements and remote-sensing-based methods (UAVs and satellite imagery) for canopy characterisation at commercial vineyard parcels. The following specific objectives were addressed:To compare the fitting results of linear regression models between manual canopy characterisation and both aerial platforms, considering different spatial and spectral resolutions.To investigate the effect of plant-by-plant versus clustered data on the precision and accuracy of canopy characteristics determination.To propose the most successful method for obtaining reliable prescription maps to be implemented in the VRA process.

## 2. Materials and Methods

### 2.1. Study Site

The research was conducted in the Alt Penedès region, one of the most important wine production areas in Catalonia, Spain. A total of five commercial vineyards (Table 1 and Figure 1) of four different varieties (Chardonnay, Merlot, Cabernet Sauvignon, and Macabeu) were included in the study. Four plots (A–D) were located in Torrelavit (Barcelona, Spain), and experiments were conducted in 2018 and 2019; the fifth plot (E) was located in El Plà del Penedès (Barcelona, Spain) and was only used for data collection in 2018. All selected vineyards were trained in a double cordon spur pruning system with green pruning when the shoot length exceeded 10 cm. All vines were in full production, non-irrigated, and with ages ranging from 21 to 31 years. The terrain slope was 5–10% for plots A–D and 0% for plot E. Furthermore, the soil was regularly harrowed to control weeds in rows and under vines.

### 2.2. Field Sampling Design

To select a representative and unbiased subset of vines and conduct the field measurements (sampling vines), a multi-stage (nested) systematic uniform random (SUR) sampling design was established. This type of sampling is more efficient than simple random sampling (with or without replacement), easy to implement, and particularly appropriate when the population is heterogeneous [44,45]. SUR sampling allows distributing the sampling locations uniformly over the entire surface of a plot, thus ensuring a known probability of selection for the entire population [46,47]. To implement multi-stage SUR sampling in each vineyard, a predefined set of sampling periods (*m*) was used to divide the entire population based on its structure. In this study, vine period refers to the number of plants between sampling vines, and row period is defined as the number of plants between the rows sampled. At each level, every *m*-th unit in the population is selected, and the position of the first sampling vine is chosen with a random start [44,46]. A random start is an integer between 1 and *m* (Figure 2). The number of sampling vines per plot was selected based on the characteristics of the plot (shape, area, length, orientation of rows, and vine spacing). The characteristics of the SUR sampling for each plot are listed in Table 2.

Each sampling location consisted of a single vine (1.2 m canopy row assigned) and was appropriately identified in each plot with a coloured tape to allow easy identification in the field during the various seasons to maintain the sampling vines at the different measurement dates. Furthermore, to identify the sampling locations subsequently from aerial images, the selected vines were physically marked on the ground (between crop rows) with two white lime marks (Figure 2) to identify the start and end of each sampled vine.

### 2.3. Manual Canopy Characterisation

Manual field measurements were conducted coincident with three different canopy stages—*beginning of flowering* (BBCH 59), *berries pea size* (BBCH 75), and *beginning of ripening* (BBCH 81)—according to the BBCH monograph. The BBCH scale is a system for a uniform coding of phenologically similar growth stages of all mono- and dicotyledonous plant species [48].

Canopy characterisation for each of the sampling vines consisted of measuring the most representative parameters (canopy height and width). Manual measurements were conducted using a regular measuring tape following the EPPO standard [49]. Each measurement included 95% of the canopy, excluding protruding branches [50]. In each sampling vine, three measurements were performed by two different surveyors. The final value was calculated from the average of the corresponding six measurements per vine. Subsequently, the leaf wall area (LWA) [51] and tree row volume (TRV) [50,52,53] were calculated, being the officially recognised parameters for pesticide dose expression [49].

### 2.4. Aerial Platforms and Multispectral Sensors Used

#### 2.4.1. UAV-Based Image Acquisition

This section presents the methodology for collecting and processing the images captured using a UAV as proposed by Campos et al. [17,42]. A UAV hexacopter (model: CondorBeta, Dronetools SL, Sevilla, Spain) loaded with a multispectral camera (model: RedEdge, Micasense, Seattle, WA, USA) flew over the vineyards. The camera was equipped with five spectral bands: red (R) centred at 668 nm with a bandwidth of 10 nm, green (G) centred at 560 nm with a bandwidth of 20 nm, blue (B) centred at 475 nm with a bandwidth of 20 nm, red edge (RE) centred at 717 nm with a bandwidth of 10 nm, and near-infrared (NIR) centred at 840 nm with a bandwidth of 40 nm. Focal length was 5.5 mm and sensor resolution 1280 × 960 pixels (width × height).

Flights were conducted 95 m above ground level at a cruise flight speed of 6 m s^−1^. Overlapping zones were adjusted to 80% in the flight sense and 60% in the transverse sense. Flights were executed on the same dates and crop stages as described for manual canopy characterisation.

From the spectral images obtained using the Micasense RedEdge, an orthophotomap with a ground sample distance of 6.48 cm pixel^−1^ was obtained. Agisoft Metashape (Agisoft LLC, St. Petersburg, Russia) was the software used for photogrammetric processes. Each orthophotomap was radiometrically calibrated using calibration plates as greyscale standards (22%, 32%, 44%, and 51% reflectance), which were placed close to the area where the UAV took off and landed. These plates were placed on the ground inside the vineyard with the objective of ensuring several frames in which the both the vine canopy and the reflectance standards are present together as the UAV flies over the defined area. From each spectral band, the 12-bit digital value in each calibration panel was extracted. A power function was used to transform each pixel in the image to its corresponding reflectance value for each of the orthophotomaps. Georeferencing of the five mosaics resulting from each spectral band of the multispectral sensor was performed using fixed ground control points in the study area. The position of the natural ground control points was accurately recorded using a global navigation satellite system with real-time kinematic (RTK) correction (model: GPS1200+, Leica Geosystems AG., Heerbrugg, Switzerland). This georeferencing process was conducted only during the first flight (BBCH 65) in each season. For the remaining flights (BBCH 75 and 81), the same ground control points were maintained. The photogrammetric calculation process yielded an RMSE of 18.62 cm.

#### 2.4.2. Satellite-Based Image Acquisition

Satellite-based images were obtained from PlanetScope (PS), a commercial constellation of nanosatellites consisting of more than 130 triple CubeSat miniature satellites (<5 kg) called as Dove (Planet Labs Inc., San Francisco, CA, USA). Although PS operates under a commercial license, many of its products are open-access for research purposes. Dove satellites are equipped with a line scanner imaging sensor with four spectral bands in the blue (455–515 nm), green (500–590 nm), red (590–670 nm), and NIR (780–860 nm) regions, providing high-resolution imagery (3 m spatial resolution) with an approximately daily revisit time. Cloud-free, orthorectified, and scaled top of atmosphere radiance level 3B images [54] were acquired from the study areas in 2018 and 2019 to maximally match the dates of manual canopy characterisation to allow comparison of the two methods. Each image from PS covered approximately 192 km^2^, and one single frame captured the entire study area on each acquisition date.

### 2.5. Image Analysis

#### 2.5.1. Canopy Vigour Map Generation

For the UAV-based imagery, the process followed by Campos et al. was utilised [17,42]. The first step for image analysis was the calculation of a vegetation index that expresses the vigour of the vines at each stage of the growth season. Several indices were considered, but given its extensive knowledge among viticulturists, and the considerable literature existing characterizing vineyards by the NDVI [17,18,19,22,25], it was finally chosen for this research. The NDVI [55] has been proven to be closely correlated to biomass development and crop stress [56,57,58]. The NDVI was calculated as a combination of the R and NIR bands (Equation (1)) (Figure 3b):(1)$$\mathrm{NDVI}=\frac{\mathrm{NIR}-R}{\mathrm{NIR}+R}$$

As the vineyards were planted in rows, vineyard-only pixels were segmented from an image by applying an NDVI threshold to eliminate the undesired elements, such as weeds, shadows, and soil. In all flights, the NDVI threshold was changed and established manually based on a visual inspection of the image. The pixels below and above the selected threshold were considered noise and classified as a ‘0′ and considered vineyard pixels and coded as ‘1’, respectively. The result was a binary mask image containing vineyard-only pixels (Figure 3c).

Combining the original NDVI images and the vineyard-only masks, the vineyard rows were masked out. In the corresponding newly created images, the non-canopy pixels became ‘0‘, whereas the vineyard canopy pixels retained their original NDVI value (Figure 3d). Subsequently, an inverse distance weighting interpolation was performed to generate a continuous NDVI map (Figure 3e), which was finally classified into homogeneous vigour areas.

Finally, for classification purposes, the interpolated NDVI images were divided into quintiles (P20, P40, P60, and P80). NDVI values lower than P20, between P20 and P80, and higher than P80 were categorised as low, medium, and high vigour, respectively. This resulted in a three-class vigour level (high, medium, and low) (Figure 3f). The above-mentioned entire process is illustrated in Figure 3.

A similar approach was followed in the case of satellite imagery, where the NDVI was calculated using bands 3 (red) and 4 (NIR) from the four-band product delivered by PS (Equation (1)). Because of the low spatial resolution (pixel size was larger than the distance between the vineyard rows), segmentation between the canopy and the background elements (weeds, shadows, and soil) was not possible. The raw NDVI images were classified into three vigour levels (high, medium, and low) following the quintile rules previously explained.

#### 2.5.2. Extraction of Information of Sampling Vines

The manually measured vines had to be identified in each orthophotomap. Therefore, a multiband RGB image was generated for each plot and flying date to enhance the visualisation of the white lime marks defining the beginning and ending of each sampling vine. A rectangular polygon guided by both lime marks (Figure 4a) was manually generated using QGIS software [59]. Combining the only-vineyard pixel mask (Figure 4b) with the polygon layer of each sampling vine, the mask was clipped, keeping only the pixels (logic 0 and 1) within the rectangles of interest (Figure 4c). Finally, a polygonisation process was performed to obtain the sampling vine-only polygon contours (Figure 4d).

From each sampling vine, the following information was obtained:Raw NDVI mean: It was calculated as the mean of all pixels contained inside the sampling vine-only polygon contour. It was obtained for the UAV-based imagery (NDVI_D_) and satellite-based imagery (NDVI_S_).Clustered vigour: Each sampling vine was assigned to a vigour class (high, medium, and low vigour) based on the three zones previously defined. This information was obtained as a categorical variable. It was determined for UAV-based imagery (C_vigour_D_) and satellite-based imagery (C_vigour_S_).Polygon-projected area (Prj_area_D_): To calculate the projected area of each sampling vine, the area of each polygon contour defining the vines was calculated using the field calculator tool in QGIS software [59]. This variable was calculated only for UAV-based imagery because the canopy and background in satellite-based imagery could not be segmented.Sampling category (Edge_pnt): Based on the geographic coordinates (ETRS89 UTM31), each sampling vine was classified as an edge point depending on its position in the plot. The sampling vines located within the inner buffer of 3 m from the plot border were considered as edge points. This information was obtained as a categorical variable (YES/NO).

### 2.6. Data Management

To conduct the planned comparisons of the different methods, an organised database was generated. Additionally, following Campos et al. [17], a new variable (NDVI_D_ × Prj_area_D_) was introduced in the database, which was obtained by combining the NDVI_D_ and the Prj_area_D_. Table 3 lists the variables included in the database.

Following the database generation, Spearman’s rank correlation [60] analysis was executed to determine the relationship between remote sensing-based information and canopy structural measurements performed in the field. The remote-sensing-based variable that correlated (higher Spearman’s ρ) the most with any of the canopy characteristics manually obtained was further selected for a deeper analysis. It is important to note that in the case of satellite imagery data, the statistical analysis was performed considering two different scenarios. In the first scenario, all the points were included regardless of their classification as edge points. The second scenario only considered data points that were not at the edge of the field plots. In the case of UAV-based imagery, where single vines can be clearly detected, the above process was not required.

The datasets of both remote sensing platforms were analysed following two different scenarios: considering every single data point as an individual value and using an aggregation (clusters) mode. The first evaluation of the obtained data was conducted considering the raw values for all single data points generated by the three different measurement procedures (UAV, satellite, and manual measurements), hereafter referred as single point data (SPD) analysis. For every single point identified, the ground-measured and remote-sensing-estimated canopy parameters were evaluated, and the potential relationships were analysed. This first proposed evaluation method allowed to a pixel-based conversion of the NDVI to any canopy parameter measured for every evaluated canopy stage and every single parcel. The remarkable discontinuity in the contiguous pixels in the maps generated from the raw pixel values (Figure 3e) is a technical limitation for the VRA of the inputs. A common technique used to solve this problem is to classify the raw values into a determined number of zones or clusters, which are treated as homogeneous management areas (Figure 3f). Furthermore, the main purpose of this research was the development of practical and useful canopy maps for the VRA of pesticides. Considering the above, the relationships among the averages of the canopy height, width, TRV, and LWA and the average remote sensing-based vegetation indices for all three different zones in every parcel classified as low, medium, and high canopy vigour zones were analysed (Figure 3f). This analysis is referred as aggregated data (AD) analysis.

Statistical analysis of all involved parameters and both proposed methods was performed to determine the potential application of linear regression. In all cases, a detailed comparison of each pair of variables was performed to obtain the most suitable linear regression model. To ensure the normality assumption, the variables were Ln-transformed. If normality was not satisfied, the linear model was rejected. This process was executed using the RStudio software [61].

## 3. Results

### 3.1. General View of Measured Parameters

The main descriptive statistical parameters obtained in the canopy characterisation are shown in Figure 5 and Figure 6. The results obtained after the manual measurements (canopy height and canopy width) and the corresponding calculated parameters (TRV and LWA) present a logical development process of the canopy with the season variation (Figure 5). Based on the data, compared to starting point BBCH 59, the canopy dimensions increased by 1.5 times up to BBCH 75. This increase was due to the rapid growth rate of the green structures occurring between bud burst and the end of flowering under normal climatic conditions. After BBCH 75, stabilisation of the canopy development was observed, and the main parameters were maintained at similar levels. The results obtained using the two aerial platforms (UAV and satellite) exhibit differences (Figure 6). At all canopy stages, the NDVI statistical ranges (difference between the maximum and minimum values) obtained with the UAV were wider than those obtained using the satellites. Amplitudes of 0.38, 0.35, and 0.61 were obtained using the UAV in the first, second, and third canopy stages, respectively, whereas the corresponding amplitudes determined using the satellites were 0.1, 0.34, and 0.27, respectively.

### 3.2. Manual Data vs. UAV Variables

#### 3.2.1. Data Correlation

To determine the spectral parameter with the most suitable correlation with any vegetative parameter (canopy height, width, TRV, or LWA), Spearman rho values were analysed. Table 4 summarises Spearman’s rho correlation matrix for the UAV variables.

The correlation values of the four canopy parameters and the NDVI ranged from 0.45 to 0.62, and those of the projected area were larger, ranging from 0.75 to 0.86. Additionally, NDVI_D_ × Prj_Area_D_ was also compared with the canopy characteristics, and the obtained values ranged from 0.79 to 0.86.

An in-depth analysis of only the best correlated parameters in Table 4 indicates that the TRV is the most remarkable canopy parameter in terms of the correlation with the information obtained using the UAV. The projected area (Prj_Area_D_) and the combination of the NDVI and the projected area (NDVI_D_ × Prj_Area_D_) are the two most remarkable parameters compared to the TRV, with a rho value of 0.86 in both cases, suggesting that both are strong correlations [62]. Additionally, the UAV-based extraction of the projected area (Prj_Area_D_) is very strongly correlated with the canopy width (rho value of 0.83). This can be expected because the projected area varies owing to the changes in the vegetation width while maintaining the canopy length equal to the plantation distance. However, this parameter still presents a very strong correlation [62] with the canopy height (rho value of 0.84). Similar results are obtained with NDVI_D_ × Prj_Area_D,_ exhibiting strong correlations with the canopy height and width (rho values of 0.82 and 0.81, respectively). The previous analysis and the main objective of this research, i.e., to determine the most remarkable relationships among the spectral parameters obtained using aerial platforms and canopy characterisation values, are considered. Accordingly, NDVI_D_ × Prj_Area_D_ is found as the most remarkable parameter. Therefore, in the following sections, detailed analysis and evaluation of this relationship are presented.

#### 3.2.2. Linear Regression Model

Considering the SPD dataset, the linear regression models between NDVI_D_ × Prj_Area_D_ and all manually measured parameters describing the canopy characteristics (canopy height, canopy width, LWA, and TRV) were evaluated. The data were Ln-transformed to ensure the normality assumption for the residues. The only variable that followed this normality assumption and yielded suitable residual plots for the model (*p* > 0.05 in the Kolmogorov–Smirnov test) was the TRV (Figure 7). Considering the results obtained after normality evaluation, for the remainder evaluated variables (canopy height, canopy width, and LWA), the intended linear regression models were rejected (*p* < 0.05 in the Kolmogorov–Smirnov test).

When the normality evaluation was performed using the AD dataset, all studied variables (canopy height, canopy width, LWA, and TRV) followed the normality assumption of the residues (*p* > 0.05 in the Kolmogorov–Smirnov test). The linear regression models built as combinations of NDVI_D_ × Prj_Area_D_ and all manually measured parameters describing the canopy characteristics yielded high coefficients of determination: R^2^ of 0.93 for the canopy height, 0.84 for the canopy width, 0.91 for the LWA, and 0.94 for the TRV (Figure 8).

As shown in Figure 8, when the data are grouped by vigour zones, the correlation values among all analysed variables are improved. Therefore, considering the potential use of this technique for the implementation of the VRA process, the model shown in Figure 8 seems the most appropriate for determining the optimal volume rate considering the canopy characteristics [8,41].

### 3.3. Manual Data vs. Satellite Variables

#### 3.3.1. Data Correlation

To evaluate the correlations between the NDVI_S_ and all vegetative parameters manually obtained (canopy height, canopy width, LWA, or TRV), Spearman’s rho correlation matrices were analysed in the case of the satellite dataset. The results of the Spearman’s rho values of the satellite variables, considering or rejecting the edge points, are listed in Table 5.

Based on the results summarised in Table 5, the spectral values are affected by the border effect. The pixels located close to the edge of a parcel seem to be contaminated by adjacent elements (mainly roads), reducing the spectral value of the pixels. However, the border effect did not impact the manually measured structural parameters of the canopy. Consequently, the spectral values obtained using the satellite imagery present a certain border effect; thus, an in-depth data analysis was conducted without edge points.

Considering similar relationships among the NDVI_S_ and the evaluated structural parameters (Table 5), all variables were included in the following linear regression analysis.

#### 3.3.2. Linear Regression Model

Considering the SPD dataset, the linear regression models between the NDVI_S_ and all manually measured parameters describing the canopy characteristics were evaluated. The data were Ln-transformed to ensure the normality assumption of the residues. The only variable that followed this normality assumption and showed suitable residual plots for the model (*p* > 0.05 in the Kolmogorov–Smirnov test) was the canopy width (Figure 9). Based on the results of the normality evaluation, for the remainder evaluated variables (canopy height, LWA, and TRV), the planned linear regression models were excluded (*p* < 0.05 in the Kolmogorov–Smirnov test).

However, when the satellite data were clustered (i.e., ADA) and the same data analysis was performed, the variables following the normality assumption of the residues (*p* > 0.05 in the Kolmogorov–Smirnov test) were the canopy height, canopy width, and TRV. For the LWA, the regression model was rejected because the normality assumption was not achieved (*p* < 0.05 in the Kolmogorov–Smirnov test). The linear regression models built as combinations of the NDVI_S_ and height, width, and TRV yielded coefficients of determination R^2^ of 0.48 for the canopy height, 0.51 for the canopy width, and 0.50 for the TRV (Figure 10).

## 4. Discussion

The three methods used to measure and characterise vine development during the season are proven to detect the different patterns associated with the vegetation growth, as observed in Figure 5 and Figure 6. From BBCH 59 to BBCH 75, a relevant increase in the parameters is detected, coinciding with the rapid shoot growth of the vines in the first stages of development. This was also observed in other studies in various crops and using different spectral sensors, such as Sentinel 2, Landsat, and RapidEye [58,63,64,65,66]. Around BBCH 75, the vine structure and the canopy architecture are modified by several management operations (i.e., shoot positioning, trimming, hedging, and leaf thinning) to maintain vegetative and fruiting balance. Consequently, there are no observable differences between BBCH 75 and BBCH 81 in the parameters assessing the canopy structure. This suggests that an alternative time for data capture can be used to better describe the rapid changes in the vegetation in the first stages after the first leaves unfold. Starting from BBCH 59, a second measurement around the end of flowering (BBCH 69) can represent the evolution of the canopy structure more realistically. In comparison, a single measurement after BBCH 75 is sufficient to point out the characteristics of the vegetation until harvesting. The latter is also confirmed when the sample distribution around the regression lines is analysed (Figure 7 and Figure 8), with a clear cluster of data points corresponding to advanced development stages (high values of the canopy height, width, LWA, and TRV). In comparison, there is a lack of points in the lower end of the x and y axes in all plots. The effect of the forcing canopy architecture in the vineyard was also observed in the temporal evolution of the LWA obtained from the field measurements in BBCH 60, 61, 69, 75, 77, 79, and 81 [42].

Remote sensing data acquired using UAV and satellite platforms were used to expedite the canopy characterisation process, which is a key procedure when determining rational PPP doses and application volumes adapted to the canopy status. However, the accuracy differed between the platforms as well as between the raw and clustered data point analysis.

First, when comparing the linear regression models, a better fitting (higher coefficient of determination) was obtained with the UAV-based data than when using the satellite information. It is also remarkable to note that at least one of the four measured variables (canopy height, canopy width, LWA, and TRV) showed significant linear regression with the remotely sensed data of both aerial platforms. UAV-based NDVI × ProjArea_D_ yielded considerably high coefficients of determination above 0.84, which presents a new area of development for UAV technology as a tool for canopy characterisation to enable VRA principles. Other studies using high-resolution NDVI maps have found significant differences between vines belonging to different vigour classes [42,65,67]; however, none of them have related or modelled the structural characteristics of vines and the NDVI values in viticulture. Ampatzidis et al. [43] obtained a strong relationship (R^2^ = 0.65) between the canopy height and the NDVI from a UAV survey of citrus groves, which have a more complex structure than vertical shoot positioned-trained vines. Satellite-based results presented significant relationships (*p* < 0.001) with the canopy height, canopy width, and TRV with coefficients of determination of approximately 0.45, reflecting the importance of ensuring sufficient spatial, spectral, and temporal resolution to reduce noise in the data. In line with this, when comparing the NDVI values from both platforms, a clear difference between the NDVI statistical ranges (difference between the highest and lowest values) was found when using UAV-based data (0.38, 0.35, and 0.61) and satellite-based data (0.1, 0.34, and 0.27, respectively). These differences were caused by the differences in the sensor spectral characteristics, spatial resolution (from a few centimetres to several meters per pixel), and spectral mixing. UAV-based imagery allows canopy segmentation and partially removes vegetation, soil, and shadow spectral mixing at the pixel level [33,58]. This discrimination is impossible when using imagery at 3 m × pixel^−1^, causing a diminishing effect on the NDVIs from vines as compared to the pure canopy spectral signals acquired from the high-resolution UAV-based images. Matese et al. [33] and Gatti et al. [65] found similar results using RapidEye satellite with a 5 m × pixel^−1^ spatial resolution (NDVI statistical ranges of 0.15 and 0.13, respectively) due to the important spectral mixing effects. Devaux et al. [66] used temporal NDVI information from Sentinel 2 satellite imagery to track vineyard growth during the season, which provided a methodology to determine the approximate dates for conducting vine structure management operations. Similarly, some consistency is found with other published results when comparing the maximum level reached by satellite-based NDVIs in the period of maximum vegetation development (July in the Northern Hemisphere). Based on the data in Figure 6, the average NDVI value for the satellite-based data is 0.27, with a maximum of 0.47. This is comparable with the per plot average NDVI reported by Devaux et al. [66] (0.4) and Gatti et al. [65] (0.44) in vineyards trained in vertical shoot positioning and grown without cover crop. Moreover, using Sentinel 2 imagery at 10 m × pixel^−1^ spatial resolution, NDVI values of 0.5 were reported by the above studies as well as by Johnson et al. [18,68]. Discrepancies between the NDVI values can be attributed to the differences in the technical specifications of the radiometric sensors used in the studies and the variation in the crop management and geographic area considered. Another important effect related to the spatial resolution of remote sensing platforms that must be considered is the ability to characterise crops in the early stages of a season when the shoot length ranges from 10 to 30 cm but the canopy density is minimal. Satellite-based imagery includes a higher proportion of unwanted elements per pixel than the area occupied by the vine canopy and require a minimum shoot and leaf development for remote assessment, as observed in the data point dispersion in Figure 9 and Figure 10. The border effect on the spectral images was detected as another factor related to the spatial resolution of the platform. Pixels close to the border zones in a parcel are considerably affected by the adjacent pixels belonging to the intended zone (mainly roads), reducing the estimated NDVI values. This fact becomes more important as the spatial resolution of the platform is increased and thus is much more important in the case of satellites than in the case of UAVs [69].

Second, a comparison of the raw point data (SPD) and clustered data (AD) demonstrated that, in general, an improvement was achieved when the samples were analysed in terms of averages based on the vigour class in the field. Although this method limited the possibility of converting a grid image into a canopy characteristic grid map (cell-by-cell), it was reliable for characterising classified maps into the three classes. From a practical perspective, this is in alignment with the method by which remote sensing companies offer canopy maps. Specifically, they perform post-processing of raw NDVI maps to yield management maps with filtered and homogeneous areas, which facilitates prescription management. Concurrently, when performing spraying applications in the variable rate mode, the machine has to change the pressure and the nozzle flow rate depending on the prescription area where it is located at each moment. Volume rate prescription areas should ensure a minimum size to avoid continuous changes in the sprayer’s operational parameters, and this is well achieved when prescription maps are clustered into two or three categories.

## 5. Conclusions

This study focuses on the development of linear regression models to predict the structural characteristics of vegetation in vineyards using aerial remote sensing. Using a rapid and cost-effective technology to monitor a canopy on a high temporal and spatial basis is key to estimate the changes in the canopy volume and density and to adapt the PPP dose with increased rationality and sustainability.

The developed methodology achieved robust characterisation (R^2^ higher than 0.84 in all cases) of the TRV, LWA, canopy height, and canopy width using the vegetation indices obtained from UAV images when the remote sensing data were classified into three vigour classes. This enables a reliable determination of the canopy characteristics, allowing the generation of PPP prescription maps defining the different vigour zones, which can be completely adapted for implementation of the VRA process [17,42]. The practical applicability of the proposed methodology is limited by the number of available maps along the season, encountered difficulties and long time required to generate the canopy vigour maps, and high price of UAV services in comparison with satellite-based options.

Satellite technology was investigated to overcome these limitations. Experiments yielded statistically significant linear relationships (R^2^ > 0.48 in all cases) between the NDVI and the canopy parameters (TRV, canopy height, and canopy width). These results, together with the higher temporal resolution and lower prices compared to those of UAVs, suggested the potential benefits of using satellite-based imagery for the VRA process based on zonal vigour variability.

Irrespective of the aerial platform evaluated (UAV or satellite) and considering the final objective of a practical implementation of the VRA of a PPP, the benefits of AD management compared with SPD evaluation were demonstrated. Pesticide distribution based on canopy vigour zones [17,70] will allow significant reduction in the use of a PPP, which is in alignment with the recently published Farm to Fork strategy [71].

Thus, future studies should focus on improving canopy characterisation considering the pixel size of satellite imagery, adaptation of field measurements for validation, and full automation of the entire process.

## Figures and Tables

**Figure 1 sensors-21-02363-f001:**
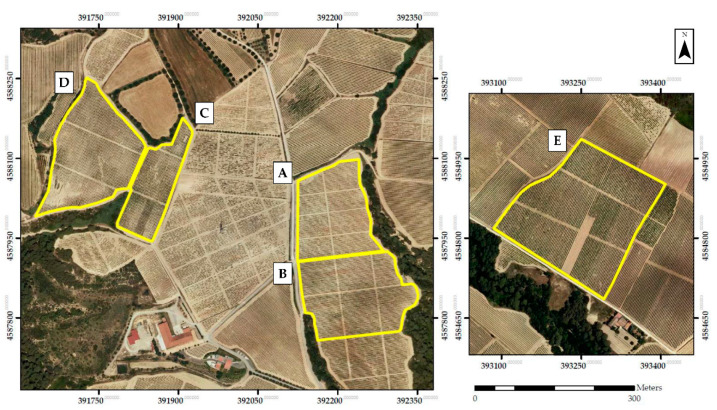
Orthophotomaps of studied vineyard plots.

**Figure 2 sensors-21-02363-f002:**
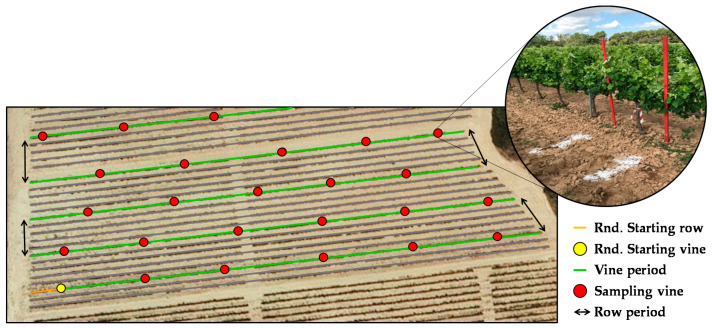
Scheme of SUR sampling used in plot A in 2018. Identification system for sampling vines in field.

**Figure 3 sensors-21-02363-f003:**
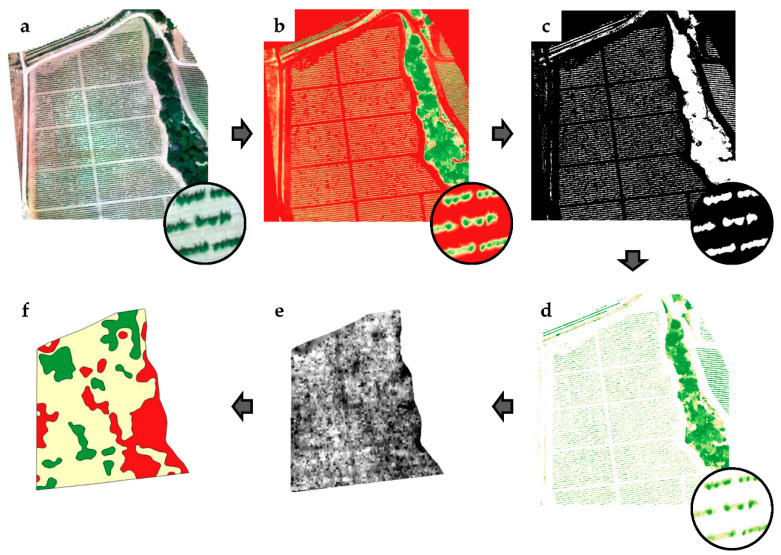
Analysis of workflow to obtain clustered vigour maps: (**a**) radiometrically calibrated multiband image, (**b**) NDVI image, (**c**) binary mask of vineyard-only pixels, (**d**) NDVI vineyard-only pixels, (**e**) continuous NDVI map, (**f**) clustered vigour map (red: low vigour, yellow: medium vigour, green: high vigour).

**Figure 4 sensors-21-02363-f004:**
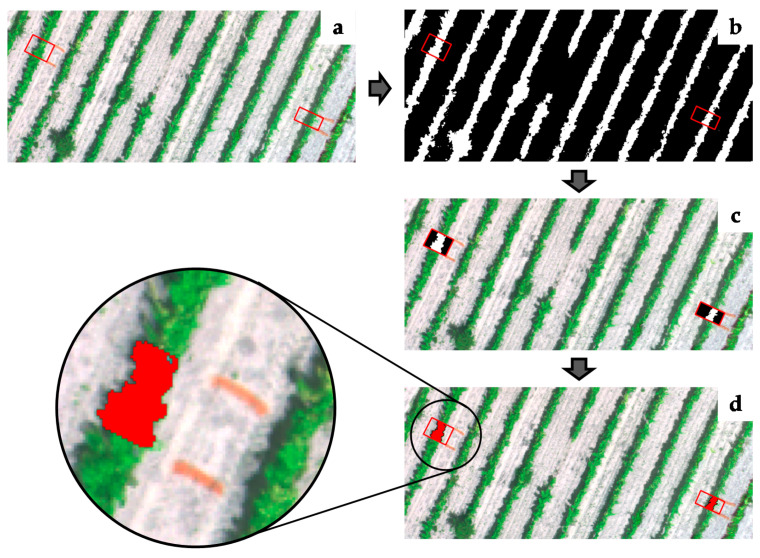
Overview of entire process from lime marks in ground to obtain sampling vine-only polygon contours: (**a**) polygons defining sampling vines, (**b**) binary mask, (**c**) sampling vine binary mask, (**d**) sampling vine-only polygon contours.

**Figure 5 sensors-21-02363-f005:**
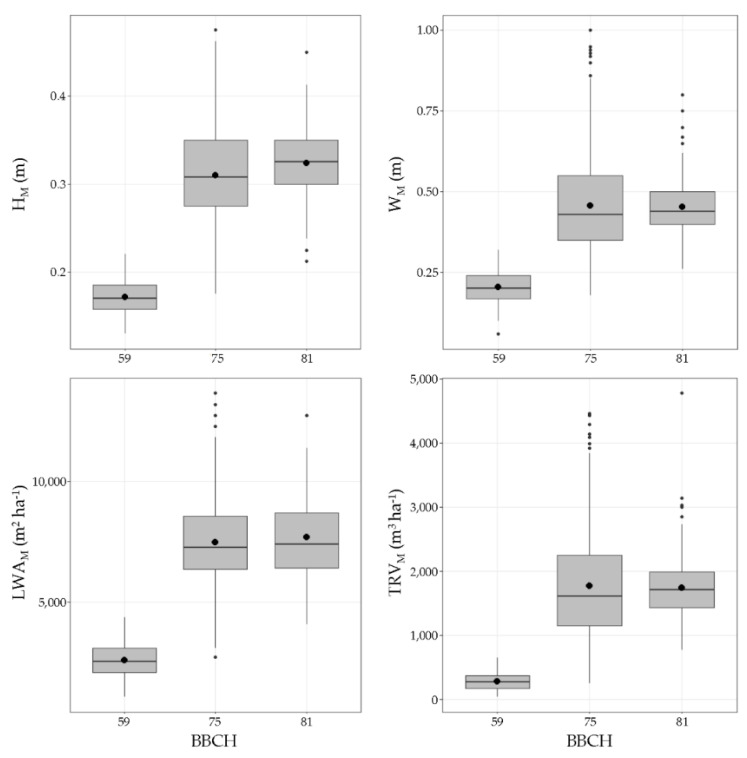
Box plots of principal descriptive statistical parameters for manual measurements. ● mean values.

**Figure 6 sensors-21-02363-f006:**
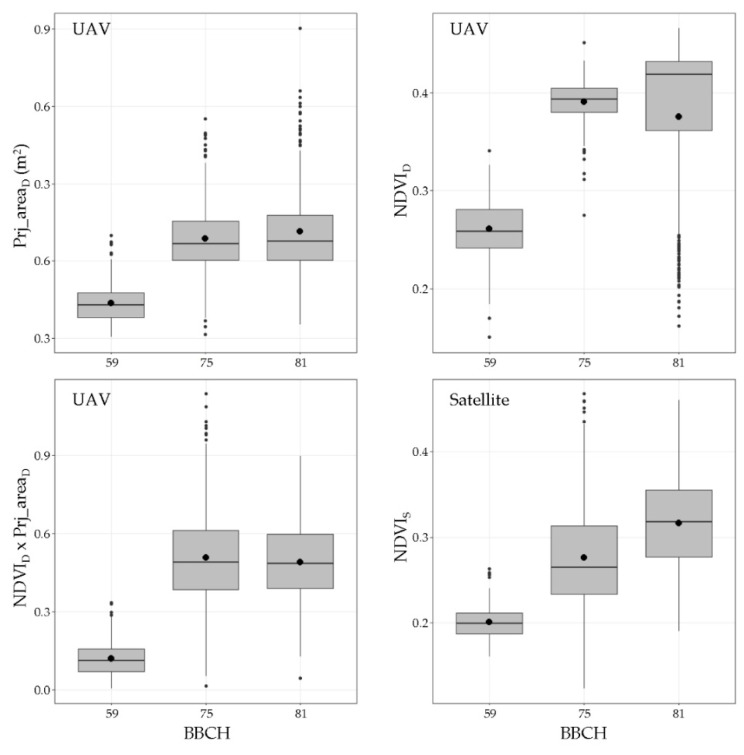
Box plots of principal descriptive statistical parameters for remote sensing variables. ● mean values.

**Figure 7 sensors-21-02363-f007:**
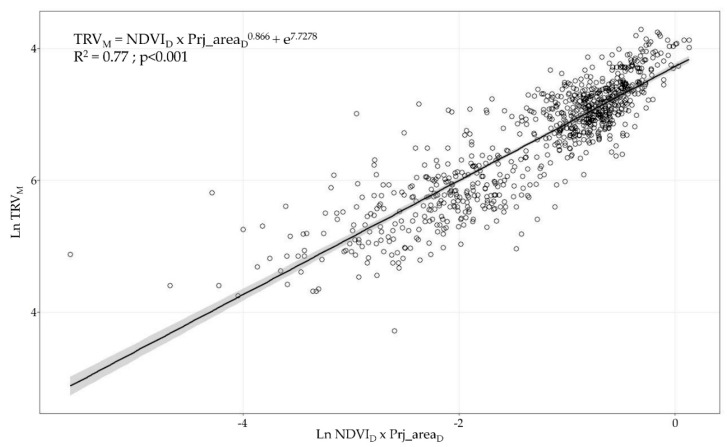
Linear regression model (R^2^ = 0.77, *p* < 0.001) evaluating relationship between Ln TRV_M_ and Ln NDVI_D_ × Prj_area_D._ Grey band shows 95% confidence interval.

**Figure 8 sensors-21-02363-f008:**
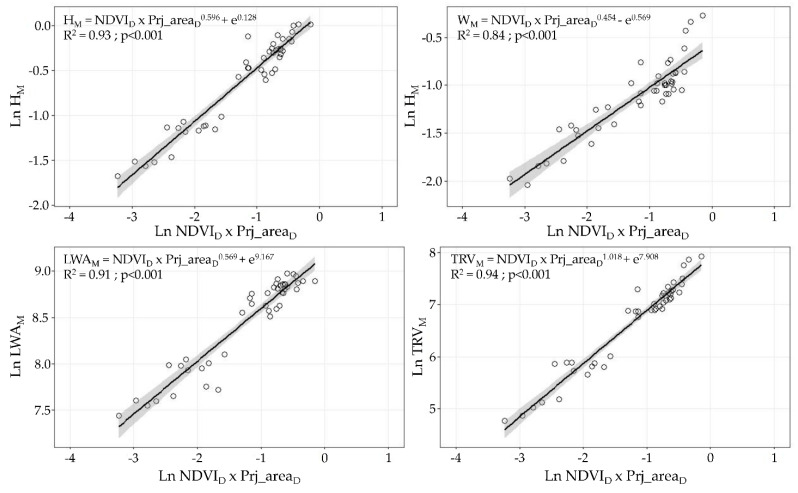
Linear regression models evaluating relationship among Ln NDVI_D_ x Prj_area_D_ and transformed canopy structural parameters (Ln H_M_, Ln W_M_, Ln LWA_M_, and Ln TRV_M_). Grey bands present 95% confidence interval.

**Figure 9 sensors-21-02363-f009:**
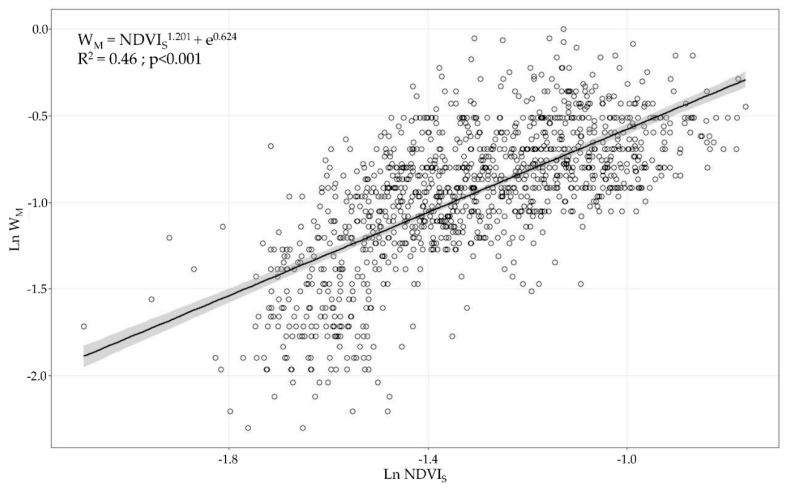
Linear regression model (R^2^ = 0.46, *p* < 0.001) evaluating relationship between Ln W_M_ and Ln NDVI_S_. Grey band shows 95% confidence interval.

**Figure 10 sensors-21-02363-f010:**
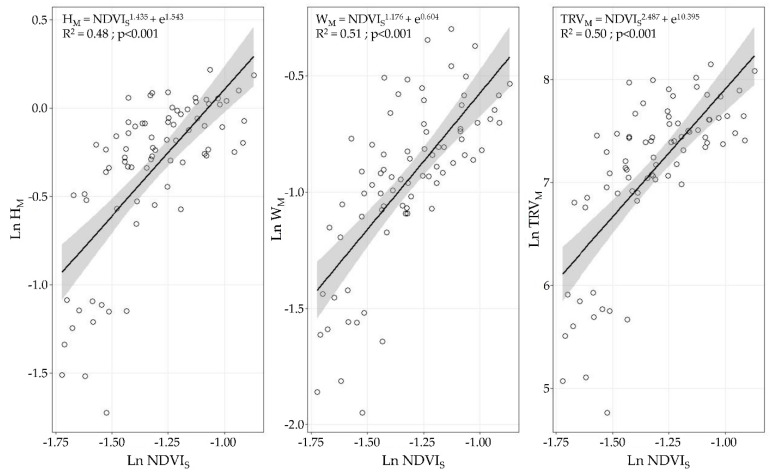
Linear regression models evaluating relationships among Ln NDVI_S_ and transformed canopy structural parameters (Ln H_M_, Ln W_M_, and Ln TRV_M_). Grey bands present 95% confidence interval.

**Table 1 sensors-21-02363-t001:** Main characteristics of selected vineyard plots.

Plot	Variety	Row Spacing (m)	Vine Spacing (m)	Area (ha)	X Coord. (m)	Y Coord. (m)	Ref. System
A	Chardonnay	2.2	1.2	2.35	392,194	4,587,999	ETRS 89 UTM31
B	Merlot	2.2	1.2	2.97	392,234	4,587,843	
C	C. Sauvignon	2.2	1.2	1.53	391,856	4,588,055	
D	C. Sauvignon	2.2	1.2	3.14	391,744	4,588,107	
E	Macabeu	2.8	1.2	4.90	391,265	4,584,841	

**Table 2 sensors-21-02363-t002:** Characteristics of SUR sampling for each vineyard plot in 2018 and 2019.

Plot	2018					2019				
	Rnd. Starting Row	Rnd. Starting Vine	Row Period	Vine Period	Total Samp. Vines	Rnd. Starting Row	Rnd. Starting Vine	Row Period	Vine Period	Total Samp. Vines
A	3	6	5	30	70	2	10	5	30	72
B	2	17	7	35	50	3	7	7	35	52
C	3	6	4	35	34	2	10	5	30	32
D	5	20	7	25	56	3	4	7	25	58
E	3	8	9	23	68	-	-	-	-	-

**Table 3 sensors-21-02363-t003:** Database fields.

Database Variables	Units	Type of Acquisition	Type of Data	Example of Data
Plot		-	Categorical	A
Vineyard variety		-	Categorical	Chardonnay
Year		-	Categorical	2019
BBCH		-	Categorical	75
Sampling vine		-	Numerical	35
NDVI_D_		UAV	Numerical	0.72
C_vigor_D_		UAV	Categorical	Medium
Prj_area_D_	m^2^	UAV	Numerical	0.17
NDVI_D_ × Prj_area_D_		UAV	Numerical	0.12
H_M_	m	Manual	Numerical	0.78
W_M_	m	Manual	Numerical	0.33
LWA_M_	m^2^ ha^−1^	Manual	Numerical	7090.91
TRV_M_	m^3^ ha^−1^	Manual	Numerical	1170.00
NDVI_S_		Satellite	Numerical	0.54
C_vigor_S_		Satellite	Categorical	Medium
Edje_pnt		Satellite	Categorical	NO

**Table 4 sensors-21-02363-t004:** Spearman’s rho correlation matrix for variables obtained using UAV.

	(1)	(2)	(3)	(4)	(5)	(6)	(7)
(1). NDVI_D_	1						
(2). Prj_area_D_	0.44	1					
(3). NDVI_D_ × Prj_area_D_	0.66	0.95	1				
(4). H_M_	0.47	0.84	0.82	1			
(5). W_M_	0.45	0.83	0.81	0.81	1		
(6). LWA_M_	0.62	0.75	0.79	0.96	0.71	1	
(7). TRV_M_	0.53	0.86	0.86	0.95	0.92	0.91	1

**Table 5 sensors-21-02363-t005:** Spearman’s rho values for variables obtained using satellite imagery considering and rejecting edge points.

	H_M_	W_M_	LWA_M_	TRV_M_
NDVI_S_ considering edge points	0.49	0.52	0.40	0.52
NDVI_S_ rejecting edge points	0.66	0.67	0.57	0.68

## Data Availability

Not applicable.

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
