# Peer review of "Assessment of Vineyard Canopy Characteristics from Vigour Maps Obtained Using UAV and Satellite Imagery"

_sensors, 2021, doi:10.3390/s21072363_

Round 1

Reviewer 1 Report

The authors prepared a well-structured manuscript. In general, it can be published after corrections.
In the introduction, a literature review should be given, what parameters are used to characterize grapes - what approaches are used and what results have been obtained. How is your approach different?
The methodology section does not indicate what photogrammetric software was used to create the orthophoto.
Also, this section does not explain why only NDVI metric was calculated, but not EVI, TVI, etc. 
The section with the results should also be corrected a bit - it is not clear whether the correlation coefficients were calculated for NDVI obtained from the UAV or satellite image. It would also be desirable to give a table of correlation of NDVI from UAV and satellite image.
Figure 9 - very similar to regression based on natural logarithm (y = ln x) rather than linear.
Basic background information should be given as to what BBCH is - not all readers may know about such scales for estimating vegetation growth.

Author Response

Reviewer #1

In the introduction, a literature review should be given, what parameters are used to characterize grapes - what approaches are used and what results have been obtained. How is your approach different?

This comment has been addressed in the introduction. Parameters as canopy height, canopy width, LWA, LAI and TRV have been deeply commented and measured during field trials. The differences between the methodologies to characterize the canopy with the proposed methods of this research has been discussed in the discussion section.

The methodology section does not indicate what photogrammetric software was used to create the orthophoto.

The comment has been addressed in line 204

Also, this section does not explain why only NDVI metric was calculated, but not EVI, TVI, etc. 

The comment has been addressed in lines 239-242. Additionally, NDVI was selected as good indicator for canopy development due its wide dissemination and use among farmers. NDVI index has experienced a large increase in their use among professionals for many other subjects linked with crop management (irrigation, fertilization, pruning…).

The section with the results should also be corrected a bit - it is not clear whether the correlation coefficients were calculated for NDVI obtained from the UAV or satellite image. It would also be desirable to give a table of correlation of NDVI from UAV and satellite image.

The results are distributed following the structure detailed below:

The results of the correlation coefficients corresponded to the NDVI obtained from the UAV are shown in the section 3.2.

The results corresponding to the NDVI obtained from the satellite are shown and discussed in the section 3.3.

The aim of this research is to find a simple method to determine canopy characteristics from different aerial platforms. For this reason, the comparison between the NDVI values obtained from the different platforms was not calculated.

Figure 9 - very similar to regression based on natural logarithm (y = ln x) rather than linear.

The objective of this paper was to evaluate the results of linear regression models between manual canopy characterization and any remote sensing variable obtained with aerial platforms (UAV and satellite)

Furthermore, it is not possible to make a regression based on natural logarithm since when the data transformation to ensure the normality assumption, negative values were obtained.

Basic background information should be given as to what BBCH is - not all readers may know about such scales for estimating vegetation growth.

Text has been modified including new explanation about the BBCH

Reviewer 2 Report

The manuscript describes the use of UAV and satellite survey techniques to determine correlations between certain geometric parameters applied to vineyards and the NDVI index and derivatives. The case study examined is a number of vineyard plantations with different morphological characteristics.
The work reports some interesting results on the correlation of the quantities analyzed.
General comments:
The abstract presents in excessive detail all the results that are discussed in the paper. I advise the authors to eliminate the linear correlation values, defining in the abstract only qualitatively the results achieved. 
The bibliography and the introduction are exhaustive for both a competent and non-competent reader and describe well the issues related to the study. However, the introduction (or at any rate the text) misses a review section on the technologies applicable to the chosen title.
For example, it is known from the bibliography to differentiate between multi-rotor UAVs and fixed-wing UAVs depending on the territorial extension and the GSD to be obtained in the orthophotos. I advise the authors to introduce these concepts.
Other comments:
Line 186-212
The Pixel size for each type of camera installed on the sensor and the focal length should be specified, so that the calculation GSD can be verified. Also, the flight parameters need to be justified: for example, the height of 95 m above ground may seem excessive on agricultural land, especially when using a multi-rotor. 
Furthermore, it is mentioned that "an orthophoto map with a ground sample distance of 6.33 cm pixel-1 was obtained". using which software application was the orthophoto map calculated? This is important because the use of one application or another will also vary the radiometric calibration algorithm (line 200-201). It would be useful to specify which calibration method was used, using which software. Also, some studies stress the importance of precision farming studies of radiometrically calibrating a pair of sensors by using the sun as the illumination source. Can the non-use of this sensor bring alterations to the study? Comment.
Generally, the use of GCP on the ground produces RMSEs from the photogrammetric calculation, which must be compatible with the scale of representation in order not to produce errors. I am sure this is verified in your work but for the sake of completeness of the reader's information, it must be included. For example, RMSE in your study must be at least less than the distance between the rows of vines, therefore below the order of magnitude of one meter.
Line 141 Figure 1: Place the geographical grid on the image, over the scalimeter

Author Response

Reviewer #2

The abstract presents in excessive detail all the results that are discussed in the paper. I advise the authors to eliminate the linear correlation values, defining in the abstract only qualitatively the results achieved. 

Text has been modified

The bibliography and the introduction are exhaustive for both a competent and non-competent reader and describe well the issues related to the study. However, the introduction (or at any rate the text) misses a review section on the technologies applicable to the chosen title. For example, it is known from the bibliography to differentiate between multi-rotor UAVs and fixed-wing UAVs depending on the territorial extension and the GSD to be obtained in the orthophotos. I advise the authors to introduce these concepts.

Text has been modified including new explanation.

Line 186-212

The Pixel size for each type of camera installed on the sensor and the focal length should be specified, so that the calculation GSD can be verified. Also, the flight parameters need to be justified: for example, the height of 95 m above ground may seem excessive on agricultural land, especially when using a multi-rotor. 

The comment has been addressed in line 198. Focal length 5,5 mm and imager size 1280x960 yielded a GSD of 6.48 m at 95 m high.

This altitude was selected since although the study was conducted in 5 plots from an 80 ha farm, a practical application was sought, and in order to increase drone performance and work load, in the future we will need to fly close to its maximum altitude permitted (near 120 m) while keeping the reliability of results. An altitude of 95 m with the type of camera used offered near 7 cm/pixel resolution which ensures enough pure pixels per vine in order to produce reliable results and calculations.

Furthermore, it is mentioned that "an orthophoto map with a ground sample distance of 6.33 cm pixel-1 was obtained". using which software application was the orthophoto map calculated? This is important because the use of one application or another will also vary the radiometric calibration algorithm (line 200-201).

The comment has been addressed in line 204

It would be useful to specify which calibration method was used, using which software. Also, some studies stress the importance of precision farming studies of radiometrically calibrating a pair of sensors by using the sun as the illumination source. Can the non-use of this sensor bring alterations to the study? Comment.

The comment has been addressed in line 210. Any GIS software allowing to performing raster calculations can be utilized to calibrate the orthophotos. The function that relates digital values (12-bit) and reflectance values (0-100%) was generated using R-software but any statistics program that allows regression analysis can be used.

It is not expected influence of light conditions in this particular study since very short flights were performed (<10 min per flight). Furthermore, it was selected clear sky days, in the summer and around noon, where sun angle varies in small degree in a 10 minute time lapse. It is true that UAV and satellite sensors should be cross-calibrated for their direct comparison, but this is not a subject for this paper and in any case the values of the two sensors have been compared between them.

Generally, the use of GCP on the ground produces RMSEs from the photogrammetric calculation, which must be compatible with the scale of representation in order not to produce errors. I am sure this is verified in your work but for the sake of completeness of the reader's information, it must be included. For example, RMSE in your study must be at least less than the distance between the rows of vines, therefore below the order of magnitude of one meter.

The comment has been addressed in line 219

Line 141 Figure 1: Place the geographical grid on the image, over the scalimeter

Figure 1 has been modified including the geographical grid